# Organic Compounds and Antibiotic-Resistant Bacteria Behavior in Greywater Treated by a Constructed Wetland

**DOI:** 10.3390/ijerph20032305

**Published:** 2023-01-28

**Authors:** Naomi Monsalves, Ana María Leiva, Gloria Gómez, Gladys Vidal

**Affiliations:** 1Environmental Engineering & Biotechnology Group (GIBA-UDEC), Environmental Science Faculty, Universidad de Concepción, Concepción 4030000, Chile; 2Water Research Center for Agriculture and Mining (CRHIAM), ANID Fondap Center, Victoria 1295, Concepción 4030000, Chile

**Keywords:** greywater, constructed wetlands, antibiotic-resistant bacteria, organic compounds

## Abstract

Laundry greywater is considered as an alternative source of non-potable water, as it is discharged in approximately 70% of homes. Because this water contains compounds such as biodegradable and recalcitrant organic matter, surfactants, and microbiological compounds, it must be treated prior to reuse. Therefore, the objective of this study was to assess the behavior of organic matter and antibiotic-resistant bacteria (ARB) in greywater treated by a constructed wetland (CW). The results show that the organic matter removal efficiencies were 67.19%, 50.15%, and 63.57% for biological oxygen demand (BOD_5_), chemical oxygen demand (COD) and total organic carbon (TOC), respectively; these efficiencies were not significant (*p* > 0.05). In addition, the CW allows the distribution of TOC and ionic compounds in the fractions below 1000 Da to increase by 5.03% and 13.05%, respectively. Meanwhile, the treatment of microbiological compounds generated non-significant removals (*p* > 0.05), along with increases in bacteria resistant to the antibiotics ciprofloxacin (CIP) and ceftriaxone (CTX) of 36.34%, and 40.79%, respectively. In addition, a strong association between ARB to CIP, CTX, cationic and non-ionic surfactants was determined, indicating the role of surfactants in ARB selection. It is suggested that disinfection systems should be employed prior to the reuse of the treated water.

## 1. Introduction

Demographic growth, urbanization, and climate change have contributed to a decrease in the quantity and quality of water sources [1,2]. Thus, almost a quarter of humanity (±1.6 billion) faces severe water scarcity, with an estimated 663 million people not having access to reliable sources, a figure that is expected to double in 10 years [3,4]. This has led to increases in mortality and morbidity associated with intestinal, respiratory, and nutritional diseases, mainly in underdeveloped countries [5,6].

To meet this growing water demand, a search has begun for alternative water sources. Greywater can be considered for non-potable purposes, as it is a resource that comprises approximately 70% of the water produced in homes [2,7]. Water from laundry accounts for a large portion of greywater and contains variable pollutant levels, depending on the load washed, products used, and water supply quality [8,9]. Salts, organic matter, and surfactants are the main components of greywater, with average abundances of 790 mg/L, 940 mg/L, and 118.3 mg/L, respectively [10,11,12]. These pollutants can have long-term effects on soil quality and cause negative impacts on both human and ecosystem health due to their high toxicity [4,12,13]. Ghaitidak and Yadav [7] and Porob et al. [14] have highlighted the presence of microbiological contaminants such as fecal coliforms and antibiotic-resistant bacteria (ARB) at concentrations of 4.0 × 10^6^ MPN/100 mL and 4.5 × 10^6^ CFU/100 mL, respectively [7,14]. These pollutants directly affect human health, as they can cause outbreaks of gastrointestinal diseases and trigger the transmission of antibiotic resistance, limiting the treatment of infectious diseases [15,16,17]. Therefore, the reuse of laundry greywater without prior treatment is not recommended for activities in which it could come in contact with humans, animals, or vegetation.

Constructed wetlands (CW) are non-conventional treatment technologies that imitate the purification capabilities of natural wetlands [18]. These systems use biodegradation as the main mechanism to effectively reduce organic matter and nutrients [19,20]. Biodegradation consists of the use of microorganisms that can assimilate contaminants to use them as substrates [21]. Greywater is considered as a biodegradable matrix because of its biological oxygen demand (BOD_5_) to chemical oxygen demand (COD) (BOD_5_/COD) ratio between 0.31 and 0.71 [22,23]. Therefore, CW treatment can be considered as a possibility for the subsequent reuse of laundry greywater [24,25]. However, the biodegradation capacity of the influents could be decreased due to the presence of surfactants and low nutrient loads [12]. In addition, it has been reported that CWs are not efficient for pathogen removal and could trigger the spread of resistance into the environment [17,26,27].

Thus, the objective of this study was to carry out an exhaustive characterization of laundry greywater, with a CW used to assess the behavior of organic compounds and their influence on antibiotic resistance. This study contributes to the knowledge about greywater, its risks and the behavior of its components when treated by a CW. This is because it includes analyses that explore the biodegradability of organic compounds and their behavior, through the application of a molecular distribution study. It also analyzes the relationship between organic compounds and the presence of antibiotic resistance in greywater, thanks to the application of a principal component analysis (PCA).

## 2. Materials and Methods

### 2.1. Description of the System and Operating Characteristics

A vertical subsurface flow (VSSF) CW was used on a laboratory scale and with semi-controlled temperatures between 1 and 20 °C on the premises of the Environmental Engineering & Biotechnology Group (GIBA-UDEC) at the School of Environmental Science, Universidad de Concepción, Concepción, Chile. The VSSFCW was constructed using acrylic with a height, width, depth, and surface area of 71 cm, 15 cm, 60 cm, and 225 cm^2^, respectively. The VSSFCW was planted with *Schoenocplectus californicus* and gravel and zeolite were used as a support medium.

The influent used to feed the VSSFCW system was greywater collected from a laundry in the city of Concepción, Biobío Region, Chile. The VSSFCW system was fed intermittently for 156 d, with daily pulses of 250 mL every 6 h. A peristaltic pump was used to feed the influent through the upper part of the system, where the greywater was filtered through the support medium by gravity. The effluent was collected through a piping system under the system and then in water drums and stored at a temperature of 5 °C. In addition, the VSSFCW system was operated at an organic loading rate (OLR) of 24.6 gCOD/m^2^d, a hydraulic loading rate (HLR) of 0.04 m^3^/m^2^d, a hydraulic retention time (HRT) of 5 d, an effective volume of 4 L, and an operating time of 156 d.

### 2.2. Analytic Methods

#### 2.2.1. Water Quality Parameters

To determine the water quality parameters for the influent and effluent of the VSSFCW, the samples were filtered with a Whatman membrane with a pore size of 0.7 µm. Then, they were analyzed based on the protocol described in the Standard Methods document [28]. Parameters such as pH, temperature (*T*), dissolved oxygen (DO), electric conductivity (EC), and oxidation–reduction potential (ORP) were measured in situ in all the samples using an OAKTON-PC650 multiparameter (Eutech Instruments; Singapore). DO was measured with the handheld oximeter oxi 330i (WTW, Xylem Analytics Germany Sales, Oberbayern, Germany).

The organic matter present in the samples was determined in the form of COD (colorimetric method, 5220-D), BOD_5_ (azide-modified Winkler method, 5210-B), total organic carbon (TOC) (catalytic combustion by oxidation and detection method NDIR, TOC analizer LCPH, Shimadzu, Kyoto, Japan).

The analyzed nutrients were measured in the form of NH_4_^+^-N, NO_3_^−^-N, NO_2_^−^-N, PO_4_^3−^-P (APHA, 2005; standard method) (spectrophotometer UV–Vis Shimadzu UV 1800, Kyoto Japan), total Kjeldahl nitrogen (TKN-N) (distillation method, digestor TT625, destilador vapodest 30s, Gerhardt, Germany), total nitrogen (TN) (Spectroquant-Nova 60, kits Merck, Darmstadt, Germany), and total phosphorus (TP) (Spectroquant-Nova, Merck kits, Darmstadt, Germany).

For the ion analysis, the samples were filtered through a 0.22 µm membrane filter. The analysis of NO_2_^−^, NO_3_^−^, F^−^, SO_4_^2−^, PO_4_^3−^, Cl^−^, and Br^−^ ions and Na^+^, NH_4_^+^, K^+^ and Li^+^ cations was conducted in accordance with the standard method using ionic chromatography (930 Compact IC Flex, Metrohm, Herisau, Switzerland).

In addition, the concentrations of three types of surfactants—anionic, cationic, and non-ionic—in the VSSFCW influent and effluent samples were measured. The analysis of surfactants such as methylene blue active substances (MBAS) was performed by means of the formation of a chloroform-extractable ion pair that was subsequently measured photometrically (Standard Method 5540-C, spectrophotometer UV–Vis Shimadzu UV 1800, Kyoto, Japan). The cationic surfactants were measured photometrically through the addition of disulfine blue anionic dye, using a cuvette test (101764, Merck Millipore, Darmstadt, Germany). Meanwhile, the non-ionic surfactants were measured photometrically by means of a reaction with the indicator tetrabromophenolphthalein ethyl ester (TBPE), using a cuvette test (101787, Merck Millipore, Darmstadt, Germany).

#### 2.2.2. Determination of Biological Contaminants

The VSSFCW influent and effluent samples were measured for biological contaminants such as fecal coliforms (FC), total coliforms (TC), and ARB, which were analyzed in accordance with the protocol described in Standard Methods [28]. FC and TC were determined by means of the multiple tube technique, using the most probable number (MPN) technique (Standard Method 9221-TC). In addition, ARB abundances were determined using the plate count technique, which is based on the ability of bacterial colonies to grow in the presence of an antibiotic. The water samples were incubated at 30 °C for 24 h using MacConckey agar as a culture medium. The plates were supplemented with the most used antibiotics in Chile as the selection media [29], including ciprofloxacin (CIP), amoxicillin (AMX), and ceftriaxone (CTX) at concentrations of 2 µg/mL, 32 µg/mL, and 4 µg/mL, respectively [30]. This technique can be used to determine ARB abundances expressed in colony-forming units (CFU)/100 mL.

#### 2.2.3. Molecular Weight Distribution

The VSSFCW influent and effluent samples underwent a molecular weight distribution analysis of TOC, anions, cations, and EC. To this end, ultrafiltration (UF) was carried out in an agitated 450 mL cell (UHP 76, Advantec MFS, Inc., Dublin, CA, USA) at 20 °C using nitrogen gas. The UF was performed using three cellulose membranes with nominal molecular weight cutoffs of 10,000, 5000, and 1000 Da, allowing the following four different fractions to be obtained: fractions with compounds greater than 10,000 Da “>10,000 Da”, compounds between 10,000 and 5000 Da “10,000–5000”, compounds between 5000 and 1000 Da “5000–1000,” and compounds under 1000 Da “<1000” [31,32]. Each of the obtained fractions underwent quantification of TOC, ions, and EC, as mentioned in Section 2.2.1.

### 2.3. Statistical Analysis

The organic matter, surfactant, ion, and microbiological removals obtained by the CW underwent statistical analysis using RStudio version 1.2.1335, with a significance level of *p* = 0.05. The Shapiro–Wilk and Fligner–Killen tests were performed to analyze normality and homogeneity of variance. Then, an ANOVA test was carried out for the data with a normal distribution and a Kruskal–Wallis test for the data without a normal distribution.

In addition, analysis of the correlation between the surfactant reductions obtained by the wetlands and the BOD_5_/COD ratio was conducted. The correlation analysis was performed using RStudio, with the “corrplot” package. A correlation factor of <0.7 was used.

A PCA was also carried out, providing a simple means of determining associations between the ARB, coliform, surfactant, and BOD_5_/COD variables. The analysis was performed with RStudio using the “vegan” package. Scores near ±1.0 indicate a strong association between the variables and principal components [33].

## 3. Results

Table 1 shows the characterization of different laundry greywater samples (*n* = 25) with respect to in situ and physicochemical parameters such as nutrients, organic matter, surfactants, and ions. Regarding the in situ parameters, the assessed greywater presented average *T*, pH, ORP, EC, DO, and turbidity values of 15.70 °C, 6.84, 70.28 mV, 660.74 µS/cm, 2.52 mg/L, and 99.70 NTU, respectively. With respect to nutrients, TN, TKN-N, forms of nitrogen, TP, and PO_4_^3−^-P were evaluated; the assessed greywater contained average concentrations of 16.64, 5.93, 0.64, 0.58, and 0.28 mg/L, respectively. Organic matter was evaluated based on COD, BOD_5_, and TOC, which occurred at average concentrations of 562.29, 296.82, and 7.23 mg/L, respectively. Anionic, cationic, and non-ionic surfactants were found at average concentrations of 18.3, 0.23, and 5.26 mg/L, respectively. With respect to the ions present in the laundry greywater, NH_4_^+^, K^+^, and Na^+^ cations were detected at average concentrations of 19.59, 5.62, and 111.34 mg/L, respectively. Meanwhile, NO_2_^−^, NO_3_^−^, F^−^, SO_4_^2−^, PO_4_^3−^, and Cl^−^ anions were detected at average concentrations of 0.14, 0.18, 3.17, 68.03, 0.27, and 398.20 mg/L, respectively.

Table 2 presents the microbiological characterization of the assessed laundry greywater. These water samples contained average TC, FC, and ARB concentrations of 70.4 × 10^6^ MPN/100 mL, 2.15 × 10^6^ MPN/100 mL, and 0.65 × 10^6^ CFU/100 mL, respectively.

Figure 1 presents the concentrations in the influents and effluents of COD, BOD_5_, and TOC and reductions obtained after the treatment of laundry greywater samples (*n* = 3) with a CW. The average COD, BOD_5_, and TOC concentrations in the influents and effluents were 579.92 and 172.72 mg/L, 173.11 and 70,56 mg/L, and 21.68 and 8.01 mg/L, respectively. In terms of the removal efficiencies of treatment by a CW, the average values achieved were 67.19%, 50.15%, and 63.57%, respectively.

Meanwhile, Figure 2a shows the concentrations of anionic, cationic, and non-ionic surfactants in the influents and effluents and reductions obtained after the treatment of greywater samples (*n* = 3) by a CW. The anionic surfactant concentrations in the influents and effluents were 18.13 and 0.05, respectively, while the cationic surfactant concentrations were 0.31 and 0.13 mg/L, respectively. The non-ionic surfactant concentrations of 4.52 and 0.74 mg/L were measured in the influent and effluent, respectively. In addition, the average anionic, cationic, and non-ionic surfactant removal efficiencies were 99.73%, 44.49%, and 84.84%, respectively. Figure 2b shows that the BOD_5_/COD ratio is positively correlated with cationic and non-ionic surfactants, while it is negatively correlated with anionic surfactants.

Figure 3 shows the concentrations of ionic compounds NH_4_^+^, Na^+^, NO_2_^−^, NO_3_^−^, Cl^−^, and SO_4_^2−^ in the influents and effluents and reductions obtained after the treatment of laundry greywater samples (*n* = 3) by a CW. In general, the concentrations of these compounds remained in the ranges between 0.16 and 1318.23 mg/L and 1.27 and 83.06 mg/L in the influent and effluent, respectively. Cl^−^ was the ionic compound that presented the greatest concentrations, with average values of 1318.23 and 1215.30 mg/L. Finally, NH_4_^+^, Na^+^, Cl^−^ and SO_4_^2−^ removal efficiencies averaged 68.84%, 21.23%, 12.37% and 33.36%, respectively. In addition, there were effluent increases for NO_2_^−^ and NO_3_^−^ of 34.36% and 80.52%, respectively.

Figure 4 shows the molecular weight distributions of TOC, anions, cations, and EC in the influents and effluents of a CW. In the effluents, there was a 5.4% increase in the lowest molecular weight fractions of TOC, ionic compounds NH_4_^+^, Na^+^, SO_4_^2−^, and EC. Meanwhile, the greatest molecular weight fraction of Cl^−^ increased by 1.4% after treatment.

Figure 5a shows the concentrations of TC and FC in the influents and effluents and reductions obtained after the treatment of greywater samples (*n* = 3) by a CW. The TC concentrations in the influents and effluents were 8.4 × 10^7^ and 7.6 × 10^5^ CFU/100 mL, respectively. Meanwhile, the FC concentrations presented values of 2.11 × 10^6^ and 1.9 × 10^4^ CFU/100 mL in the influents and effluents, respectively. This treatment achieves an average removal efficiency of 82.38% for both parameters. Figure 5b shows the rate of resistance to the antibiotics CIP, CTX, and AMX of the ARB in the samples, for which the CW treatment achieved removal efficiencies of −36.34%, −40.79%, and 82.49%, respectively. Negative efficiencies indicate concentration increases in the effluents.

Figure 6 shows the PCA of the reductions in different parameters obtained after treatment by a CW. Dimension 1 (Dim 1) contributes 78.54% of the variance and presents a strong association with the reductions in bacteria resistant to the antibiotics CIP and CTX, TC, FC, non-ionic and cationic surfactants, and the BOD_5_/COD ratio. Meanwhile, there is no association between these components and the bacteria resistant to AMX and anionic surfactants.

## 4. Discussion

### 4.1. Characterization of Laundry Greywater

The values shown in Table 1 indicate that in general, the parameters evaluated for the sampled greywater are within the range reported by other authors [7,8,34]. Similarly, parameters such as ORP, EC, COD, and BOD_5_ presented high standard deviations, which was expected, as they depend on the load washed, which is different in each sampling [1]. The average pH in the assessed samples was 6.8, a value that is considered to be low, given that the pH of laundry water is usually between 7.5 and 10 [10]. With respect to nutrients, the phosphorous concentrations are considered as low (0.1–5.6 mg/L) compared to the 51.6 mg/L in laundry water reported by Ghaitidak et al. [7]. This is associated with the composition of the products used for washing, as some are free of this compound to avoid damage to the environment [1]. Nitrogen content, meanwhile, was detected at concentrations up to 78 mg/L, much higher than the 40 mg/L reported by Braga et al. [8] in laundry water. Greywater is considered to have low nitrogen concentrations because more than 90% of nitrogen originates from urine [35]. However, when evaluating the forms of nitrogen, organic nitrogen accounts for 88.9% of the nitrogen contained in the sample, which is consistent with the findings of Henderson et al. [36]. Therefore, these concentrations could be explained by the presence of surfactants and the microbiological content of the samples (Table 2), which could arise from proteins, nucleic acids, purines, and pyrimidines [37].

The organic matter concentrations detected in the laundry samples are within the ranges reported in other studies on laundry water, in which the concentrations vary between 58 and 4474 mg/L COD [8,38], 44.3 and 6250.0 mg/L BOD_5_ [7,33], and 23.0 and 36.4 mg/L TOC [24]. Thus, the organic matter content in the samples depends on lifestyle, family structure, and the products used [39]. It is worth mentioning the predominance of COD over BOD_5_, which is due to the presence of xenobiotic organic compounds (XOC), which are mainly surfactants that are not easily biodegradable [9,28].

Among the main components of greywater are surfactants, which are divided into the following different classes depending on their chemical nature: anionic, cationic, and non-ionic. The concentrations of the surfactant classes are consistent with those described in the literature, in which the most abundant are usually anionic, followed by non-ionic surfactants and, finally, cationic surfactants in lower concentrations [12,13]. Fedeila et al. [40] highlight the environmental risk of releasing effluents with anionic surfactant concentrations greater than 13.1 mg/L. Laundry greywater can also contain various ionic compounds at concentrations from <LD to 789.2 mg/L, which explains the variation in EC from 177.1 to 1704.0 µS/cm. Among the ions, Na^+^ and Cl^−^ stand out because they reach average concentrations of 111.3 and 398.2 mg/L, respectively. These concentrations are thought to occur due to their presence in soaps and detergents [7], their use as ion exchangers to soften water [41], and/or the use of Cl^−^ and F^−^ in water purification plants [42].

Table 2 also shows that the greywater is not free of coliform contamination, as TC and FC were detected at average concentrations of 70.4 × 10^6^ and 2.2 × 10^6^ MNP/100 mL, respectively. These results are consistent with those reported in studies on laundry water by authors such as Shreya et al. [34] and Ghaitidak et al. [7], who found TC and FC concentrations of 2.1 × 10^6^ and 9.5 × 10^6^ MNP/100 mL, respectively. The presence of coliforms in laundry water is attributed to personal hygiene and diaper washing [9,34]. Meanwhile ARB were detected at levels from <LD to 1.85 × 10^6^ CFU/mL. Porob et al. [14] and Craddock et al. [43] also reported ARB and antibiotic resistance genes (ARG) in greywater, with resistance rates of 11.4–69.3% in bacterial isolates. The origin of these ARB is attributed to contamination by coliforms, which can contain ARG [44], as well as the presence of selection agents, such as trace antibiotics and biocides, which can trigger the expression of resistance phenotypes [42,45,46]. The existence of biofilms in different parts of conventional washing machines has even been reported, which could also contribute to the bacterial resistance response [47].

### 4.2. Behavior of Organic Compounds in a CW

Figure 1 shows the organic matter removal efficiencies in terms of COD, BOD_5_, and TOC, which were 67.2%, 50.2%, and 63.6%, respectively. These values are associated with both aerobic and anaerobic microbial activity [34]. However, these removals were not significant (*p* > 0.05), contradicting the studies of Sotiropoulou et al. [48], Kotsia et al. [49], and Ramprasad et al. [24] on VSSFCW that treat greywater, which reported significant removals of 93% of COD, 99% of BOD_5_, and 95% of TOC, respectively.

The biodegradation capacity of the influents can be assessed by means of ratios such as BOD_5_/COD and COD/BOD_5_, the recommended values of which are 0.31–0.71 and <3, respectively [9,39]. The influents in this study presented values of 0.3 and 3.35, indicating the biodegradable nature of the influents, even when they are near the limit. Furthermore, it indicates the presence of compounds refractory to biological degradation, which are xenobiotic compounds (XOC), such as whiteners, surfactants, softeners, or beauty products [34,50]. Therefore, it can be inferred that they diminish the biodegradation capacity of greywater, as stated by Khalil et al. [12]. Similar values were reported by Ramprasad et al. [24], who found BOD_5_/COD ratios in greywater in the range of 0.29 to 0.41.

The BOD_5_:N:P ratio was used to evaluate the feasibility of treating the influents using aerobic treatment; its recommended values are 100:5:1 [51]. The values obtained for the assessed influents were 100:4.7:1, suggesting that laundry greywater has both biodegradable and refractory portions and that the nutrient content is adequate to decrease the biodegradable organic matter present [34].

The BOD_5_ concentrations in the effluents are, on average, 70.0 ± 14.3 mg/L. A review conducted by Oh et al. [52] on the quality standards for the reuse of treated greywater in different countries indicates that the obtained effluents are not recommended for reuse, as BOD_5_ concentrations of <20 and <10 mg/L, depending on the country, are mentioned. For example, for toilet flushing in New South Whales, Italy, and Israel, the concentration must be <20 mg/L, while in the USA, it must be <10 mg/L. The concentrations found in this study could be due to the origin of the influents, as, because they are found in laundry, the concentrations of both biodegradable and recalcitrant organic compounds are high. Therefore, it is suggested that mixed greywater should be used to avoid the discharge of high, constant concentrations of organic matter into soil, thereby avoiding the induction of hydrophobicity that causes a reduction in water infiltration rates and increases in surface runoff and soil erosion [1,11].

Surfactants are among the most abundant organic contaminants in laundry greywater. Figure 2a shows the surfactant removal efficiencies. The mechanisms involved in this removal are thought to be mainly adsorption, biodegradation, and phytodegradation, depending on the nature of the surfactant [10,12]. Anionic surfactants were removed at significant rates (*p* < 0.05) and presented the greatest removal efficiency, at 99.7%. Similar results were obtained by Shreya et al. [34] and Ghaitidak et al. [7], who reported efficiencies of 92–95% and 92%, respectively. These efficiencies are associated mainly with adsorption mechanisms, as due to their chain lengths, anionic surfactants can be adsorbed by substrates with greater pore sizes [10,34]. Non-ionic surfactants, meanwhile, were removed at an average rate of 84.8%, greater than that reported by Ramprasad et al. [24], who obtained efficiencies of 44%. This removal is associated mainly with biodegradation, as their hydrophilic nature and simple structure allow them to be more available for use by microorganisms [10]. Finally, cationic surfactants presented the lowest removal from the influents; however, their reduction may be associated with mechanisms such as adsorption, phytodegradation, and biodegradation [12,41]. Ramprasad et al. [25] reported a 58% removal of TMA, a cationic surfactant. This rate is related to its adsorption by substrates and obligate microbial degradation.

All the analyzed surfactants accounted for 5% of the total COD of the influents, a value below the 15% contribution reported by Hernandez et al. [22]. However, the obtained effluents contain a significant portion of recalcitrant organic matter; therefore, more studies on the XOCs present in laundry greywater are suggested, as their persistence can pose risks for subsequent reuse [53].

Figure 2b, meanwhile, is a representation of the surfactant removal mechanisms in the CW. The positive correlations (0.97) between non-ionic and cationic surfactants and the BOD_5_/COD ratio indicate that the removal mechanisms of these surfactants are associated with biodegradation processes. Meanwhile, the negative correlation (−0.78) between anionic surfactants and the BOD_5_/COD ratio could be an indication that the biodegradable organic matter of the influents increases the bacterial biofilm in the medium, generating greater filtration, and therefore greater removal. The concentrations of anionic, cationic, and non-ionic surfactants in the effluents were 0.03, 0.05, and 0.61 mg/L, respectively. Shreya et al. [34] and Hernandez et al. [22] indicate that anionic surfactant concentrations over 30 mg/L and cationic and non-ionic surfactant concentrations over 10 mg/L can have a toxic effect on microbial communities, fish, and humans [12,13].

The organic compounds that enter the CW have different origins; therefore, there are different ionic compounds at different concentrations. Figure 3 shows the ions evaluated and their behavior after being treated by a HC, where Figure 3a shows the action of the HC on the ionic compounds and Figure 3b shows the removal efficiencies. This Figure shows that the ionic compounds that contain NO_2_^−^ and NO_3_^−^ increase in the effluents, while those containing Cl^−^, Na^+^, SO_4_^2−^, NH_4_^+^ decrease. The increases in NO_2_^−^ and NO_3_^−^ ions and the reduction in NH_4_^+^ are an expected result, as within the system, nitrification processes occur due to the aerobic conditions maintained in the VSSFCW [54]. The nitrification process occurs in two oxidation stages; in the first, NH_4_^+^ is converted to NO_2_^−^ by chemolithotrophic bacteria, while in the second, NO_2_^−^ is transformed into NO_3_^−^ [55]. The organic compounds that contain Cl^−^, Na^+^, and SO_4_^2−^ can be part of surfactants, biocides, or cell debris [2], which, depending on the compound that it contains, can be reduced by mechanisms such as adsorption and biodegradation.

Figure 4 shows how these compounds enter the CW and undergo a series of transformations in which they are degraded from compounds of a greater molecular weight to other simpler compounds [11]. TOC in the influents is distributed mainly in the fraction greater than 10,000 Da, which accounts for 46.8%, while in the effluents, it is mainly in the fraction below 1000 Da, which accounts for 35%. This demonstrates the previously described phenomenon, in which primarily autotrophic bacteria use organic matter as an energy and carbon source for cell production [56], and organic matter of a high molecular weight greater than 10,000 Da decomposes into organic compounds of a low molecular weight below 1000 Da that are more easily mineralizable [36,57]. TOC in greywater can be found in the form of high-molecular-weight compounds, mainly from microorganisms, such as biopolymers and extracellular polymeric substances [58]. 

With respect to ionic compounds, NH_4_^+^ and Na^+^ are distributed mainly in the fraction below 1000 Da in both the influents and effluents. NH_4_^+^ in this fraction accounts for 63.6% and 100% of its total, respectively, and Na^+^ for 64.2% and 66.9%, respectively, indicating that the compounds that contain them are degraded during treatment by the CW, giving rise to simpler molecules. This also explains the result obtained for anionic compounds that contain SO_4_^2−^, which are distributed mainly in the 1000–5000 Da fraction, which accounts for 65.3% of its total in the influents and 100% in the effluents. Cl^−^, meanwhile, is distributed mainly in the fraction below 1000 Da in both influents and effluents, which accounts for 70.9% and 69.7% of its total, respectively. Considering that Cl^−^ decreases in the effluents, it can be assumed that the compounds that contain Cl^−^ bind to others. This can cause the formation of disinfection byproducts that react with the dissolved organic matter in the system and can generate trihalomethanes and other organochlorine compounds [59,60].

EC is distributed mainly in the <1000 Da fraction in both influents and effluents, which accounts for 52.2% and 64.7% of its total, respectively. This is an expected result, as in the low-molecular-weight fractions, there is a greater accumulation of the assessed ions after treatment by a CW.

### 4.3. Behavior of Microbiological Compounds in a CW and Their Relationship with Organic Compounds Present in Laundry Greywater

Figure 5a presents the coliform behavior in the CW. The results show that this treatment has removal efficiencies of 82.4% for both coliform types (TC and FC), which is associated mainly with adsorption, filtration, competition, antibiosis, natural extinction, and predation mechanisms [18,61,62]. Goncalves et al. [50] also reported limited removal efficiencies for microbiological compounds when using a VSSFCW to treat greywater, obtaining effluents with coliform concentrations of 100–1000 MPN/100 mL. The effluents assessed in this study contain TC and FC concentrations of 7.6 × 10^5^ and 1.9 × 10^4^ MPN/100 mL. This is consistent with the findings of various authors, who state that CW systems are not designed for the removal of *E. coli*, coliforms, or helminth eggs [9,26,50]. Because bacteria require more than 10^3^ cells to cause an infection [59], the fecal load present in the effluents can pose a health risk in any projected reuse without prior disinfection treatment [50,63]. The greatest problem is the possible aerosols generated that can cause infection through inhalation, ingestion, or contact with skin [63]. In addition, Shi et al. [58] report the presence of enteric viruses in greywater that, even at low concentrations, can cause infections.

Figure 5b, meanwhile, presents further evidence of the inefficiency of the CW at eliminating microbiological compounds, with ARB removal efficiencies of −36.3%, −40.8%, and 82.5% for CIP, CTX, and AMX, respectively. These results show that the CW generates an increase in bacteria resistant to the antibiotics CIP and CTX. Similar results were found by Chen et al. [64], He et al. [65], and Huang et al. [66], who assessed the performance of a CW related to wastewater treatment. These authors showed that the biological processes in CWs can cause the transmission and spread of resistance, increasing ARG abundance.

As the method used for ARB counting is specific for enterobacteria, the detected ARB belong to the genera *Escherichia, Salmonella, Enterococcus, Shigella,* and *Klebsiella* [43,67]. Noman et al. [44] reported a resistance rate of 10.9% among *Salmonella* isolates in greywater, while Craddock et al. [43] reported that resistance to CIP among *E. coli* isolates in greywater ranged between 9% and 10% and that 96.7% of *Klebsiella* isolates were resistant to ampicillin, a penicillin-class antibiotic similar to CTX. Therefore, the reuse of the obtained effluents could trigger intestinal diseases such as cholera, salmonellosis, or typhoid fever that are resistant to conventional antibiotic treatments [59].

Figure 6 shows the PCA, where the principal components (PC) PC1 and PC2 explain 100% of the total accumulated variance. PC1 stands out as contributing 78% of the variance and reveals a strong association with non-ionic surfactants, cationic surfactants, ARB to CTX and CIP, CT, CF and the biodegradability ratio BOD_5_/COD, with coefficients of 0.31, 0.37, 0.37, 0.37, 0.37, 0.36 and 0.36, respectively. This analysis shows the behavior of microbiological compounds, in which the ARB and coliform ratio indicates that by increasing the removal of coliforms from greywater, the ARB removal efficiencies would increase [62,68,69]. The relation between ARB and cationic surfactants is explained by the fact that they are widely used as disinfectants, as they contain biocide compounds such as quaternary ammonium (QAC) or benzalkonium chloride (BAC) [70]. These compounds can select ARB by means of co-selection or generate resistance in previously susceptible bacteria by generating point mutations [63]. Jia et al. [45] reported that *E. coli* tends to acquire resistance through genetic mutations during exposure to QAC. Regarding the relation with non-ionic surfactants, Zaky et al. [71] report that these exert a biocide action on various microorganisms, as they modify membrane structure and permeability. Therefore, they could also select resistance phenotypes among the coliforms present. Meanwhile, the relation between ARB and BOD_5_/COD could be because the greater availability of biodegradable organic matter will allow the greater spread of ARB in the medium [62]. Finally, due to their structure, non-ionic surfactants contribute more to biodegradable organic matter, which is in line with what was reported by Ramprasad et al. [10].

Due to the results obtained, studies on how to optimize the HC based on its design and operating characteristics are suggested. In order to evaluate how different conditions influence the type of microorganisms present in the HC, their behavior and capacity to degrade organic compounds must be analyzed. In addition, the use of a disinfection system to reduce the presence of TC, CF and ARB in the effluents is suggested.

## 5. Conclusions

This study allowed laundry greywater to be characterized and the performance of a CW related to organic and microbiological contaminants to be determined. The main conclusions are as follows:The organic matter removal efficiencies using a CW were 67.19%, 50.15%, and 63.57% for COD, BOD_5_, and TOC, respectively. These efficiencies were not significant (*p* > 0.05); therefore, further studies on the XOCs present in laundry greywater are suggested to evaluate their effect on biodegradation.The assessment of molecular weight distribution showed that the TOC percentage increased by 5.4% in the fraction below 1000 Da in the effluents. The same behavior was found for the ionic compounds NH_4_^+^, Na^+^, and EC, with increases of 36.44%, 2.71%, and 12.51%, respectively. Thus, the CW reduces the molecular weight of organic compounds, making them simpler, and therefore more mineralizable.The TC, FC, and ARB removal efficiencies were 82.38%, 82.38%, and 1.78%, respectively; these efficiencies were not significant (*p* > 0.05). There were also increases in the bacteria resistant to the antibiotics CIP and CTX of 36.34% and 40.79%, respectively. A strong association between ARB and CTX, CIP, cationic and non-ionic surfactants was found according to PCA. This behavior indicates the role of surfactants in resistance selection. Due to the results obtained, the use of disinfection systems is suggested to decrease the impact of treated laundry greywater on the environment and health.

## Figures and Tables

**Figure 1 ijerph-20-02305-f001:**
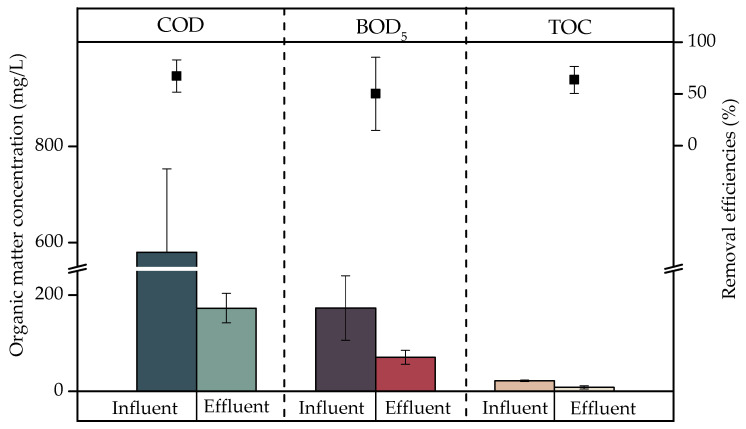
Concentrations of COD, BOD_5_, and TOC present in laundry greywater after treatment by a CW. *N* = 3.

**Figure 2 ijerph-20-02305-f002:**
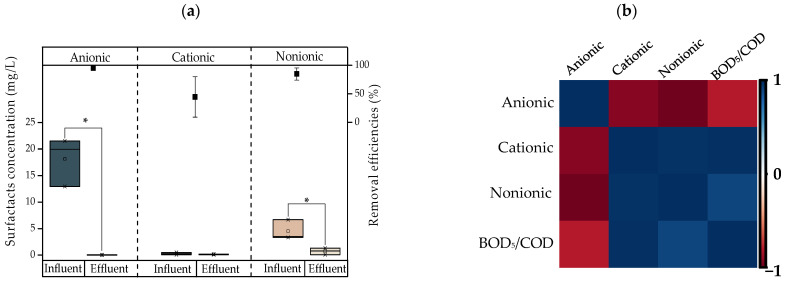
Surfactant behavior in a CW. (**a**) Concentrations of anionic, cationic, and non-ionic surfactant present in laundry greywater and removal efficiencies after treatment by a CW. *: significant reduction (*p* < 0.05). (**b**) Correlation analysis of anionic, cationic, and non-ionic surfactants and BOD_5_/COD ratio. Positive correlations between parameters are shown in blue color close to 1, while negative correlations are shown in red color close to −1.

**Figure 3 ijerph-20-02305-f003:**
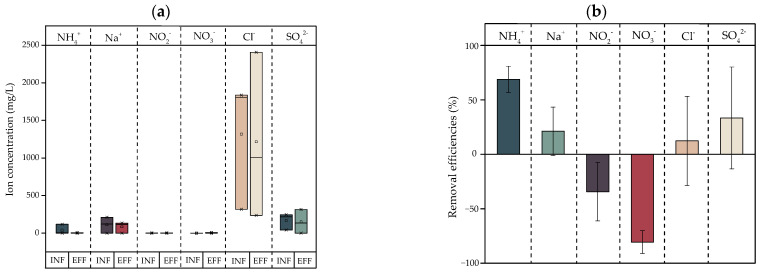
Behavior of ionic compounds in a CW. (**a**) Cationic compound (NH_4_^+^ and Na^+^) and anionic compound (NO_2_^−^, NO_3_^−^, Cl^−^ and SO_4_^2−^) concentrations in laundry greywater; (**b**) removal efficiencies (%) efficiencies after treatment by a CW.

**Figure 4 ijerph-20-02305-f004:**
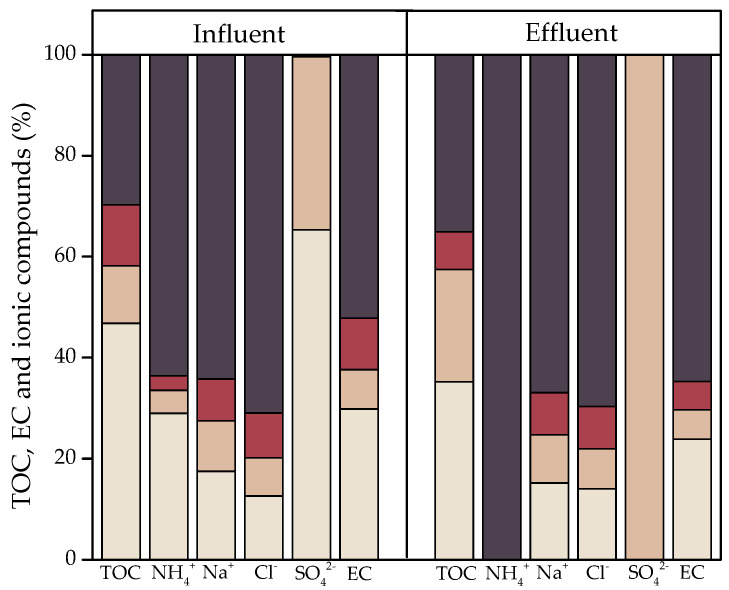
Molecular weight distributions of ionic compounds, TOC, and EC in laundry greywater before and after treatment by a CW. 
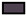
: <1000 Da; 
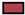
: 5000–1000 Da; 
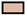
: 1000–10,000 Da; 
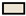
: >10,000 Da.

**Figure 5 ijerph-20-02305-f005:**
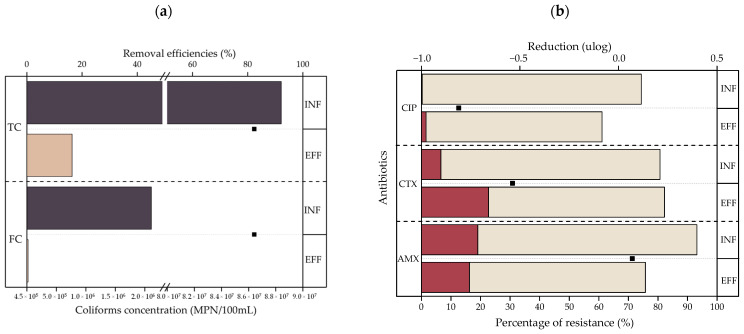
Behavior of microbiological contaminants in laundry greywater samples before and after treatment by a CW. (**a**) TC and FC concentrations in influents and effluents of a CW (bar graph) and their removal efficiencies (dot chart); (**b**) ARB rates of resistance to the antibiotics CIP, CTX, and AMX in influents and effluents of a CW (stacked bar chart), and their corresponding removal efficiencies (dot chart). 
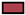
: ARB; 
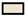
: antibiotic-susceptible bacteria.

**Figure 6 ijerph-20-02305-f006:**
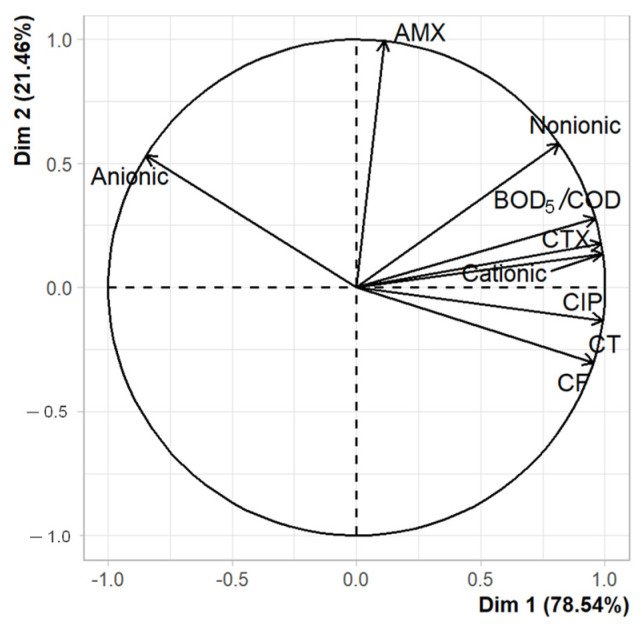
Principal component analysis (PCA) of the reductions in bacteria resistant to the antibiotics AMX, CIP, and CTX, TC and FC, anionic, cationic, and non-ionic surfactants, and the BOD_5_/COD ratio obtained after treatment by a CW.

**Table 1 ijerph-20-02305-t001:** Typical concentrations of in situ and physicochemical parameters found in laundry greywater.

Parameter		Unit	Average	Range
In situ	*T*	°C	15.7 ± 1.4	13.3–16.8
	pH	-	6.84 ± 0.35	6.58–7.54
	ORP	mV	70.28 ± 79.18	−119.70–66.60
	EC	µS/cm	660.7 ± 599.2	177.1–1704.0
	DO	mg/L	2.5 ± 1.2	1.0–4.9
	Turbidity	NTU	99.7 ± 62.2	3.1–144.0
Nutrients	NH_4_^+^-N	mg/L	0.62 ± 0.38	0.13–0.99
	NO_2_^-^-N	mg/L	0.04 ± 0.04	0.01–0.13
	NO_3_^-^-N	mg/L	1.27 ± 0.92	0.06–2.39
	TKN-N	mg/L	5.93 ± 4.03	3.29–10.56
	PO_4_^3-^-P	mg/L	0.280 ± 0.350	0.003–0.915
	TP	mg/L	0.6 ± 0.4	0.1–1.4
	TN	mg/L	16.2 ± 27.5	0.8–78.0
Organic matter	COD	mg/L	562.3 ± 279.4	67.0–902.9
	BOD_5_	mg/L	296.8 ± 387.9	3.8–940.0
	TOC	mg/L	7.23 ± 0.55	6.69–7.78
Surfactants	Anionic	mg/L	18.30 ± 4.57	12.93–32.52
	Cationic	mg/L	0.23 ± 0.18	0.11–0.49
	Non-ionic	mg/L	5.26 ± 2.14	3.34–7.50
Cations	NH_4_^+^	mg/L	19.59 ± 25.03	<LD–48.50
	K^+^	mg/L	5.62 ± 4.51	<LD–9.42
	Na^+^	mg/L	111.34 ± 106.13	<LD–211.36
	Li^+^	mg/L	<LD	<LD
Anions	NO_2_^−^	mg/L	0.14 ± 0.20	<LD–0.28
	NO_3_^−^	mg/L	0.18 ± 0.04	0.13–0.21
	F^−^	mg/L	3.17 ± 5.42	<LD–9.42
	SO_4_^3−^	mg/L	68.03 ± 61.81	26.06–139.02
	PO_4_^3−^	mg/L	0.27 ± 0.15	0.11–0.39
	Cl^−^	mg/L	398.20 ± 354.46	97.82–789.15
	Br^−^	mg/L	<LD	<LD

Note: T: temperature; ORP: oxidation–reduction potential; EC: electric conductivity; DO: dissolved oxygen; TKN-N: total Kjeldahl nitrogen; TP: total phosphorus; TN: total nitrogen; COD: chemical oxygen demand; BOD_5_: biological oxygen demand; NTU: nephelometric turbidity unit; <LD: under limit of detection (<0.01 mg/L). *n* = 25.

**Table 2 ijerph-20-02305-t002:** Concentrations of biological contaminants that can be found in laundry greywater.

Parameter		Unit	Average (×10^6^)	Range (×10^6^)
Microbiological	TC	MPN/100 mL	70.4 ± 71.0	0.7–160.0
	FC	MPN/100 mL	2.15 ± 1.63	0.017–4.0
	ARB	CFU/100 mL	0.65 ± 0.92	<LD–1.85

Note: FC: fecal coliforms; TC: total coliforms; ARB: antibiotic-resistant bacteria; MPN/100 mL: most probable number/100 mL; CFU/100 mL: colony-forming units/mL.

## Data Availability

The data presented in this study were obtained through laboratory analyses and they were not available in public databases.

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
