# Peer review of "Organic Compounds and Antibiotic-Resistant Bacteria Behavior in Greywater Treated by a Constructed Wetland"

_ijerph, 2023, doi:10.3390/ijerph20032305_

Round 1
Reviewer 1 Report
Wetland waste water treatment is an economical and environment friendly method. In this research, a lab scale constructed wetland (CW) experimental apparatus was used in treatment of laundry greywater, and the results obtained showed useful information for the readers of this field. My major suggestion is that wetland is an ecosystem, the function of the wetland is heavily dependent on the species diversity and balance of the ecosystem. In this experiment, the CW planted with Schoenocplectus californicus is too simple. More kinds of plants, microorganisms, or even animal planktons can be considered and tried. The organisms that have high degradation ability to the waste compositions contained in the waste water can be screened, enriched and/or bred and used in the CW ecosystem to improve the treatment efficiency. It should be discussed in the discussion section at least.
In the sub-title of 4.3 and some other places, ”microbiological compounds” changes to “microorganisms”.
Author Response
Dear Reviewer 1,
The answers of your comments are in the attached file.
Thank you in advance.
Best regards,
Glaadys Vidal

Reviewer 2 Report
“Organic Compounds and Antibiotic-Resistant Bacteria Behavior in Greywater Treated by a Constructed Wetland” by Monsalves et al. is an original scientific paper that examined the behavior of organic compounds and antibiotic-resistant bacteria in treated greywater. The reviewer has additional comments:
· In the last decade, many papers have been published dealing with the problem of greywater. Therefore, the authors need to emphasise more clearly the novelty of their research.
· Abstract. “along with increases in bacteria resistant to the antibiotics ciprofloxacin (CIP) and ceftriaxone (CTX) of -36.34%, and -40.79%, respectively.” Since you mention an increase, please do not use minuses to express this.
· Introduction. Use [11-13] and [16-18] instead [11, 12, 13] and [16, 17, 18], respectively.
· Chapter 2.1. Please supplement Universidad de Concepción with further information: Concepción, Chile.
· In many places in the paper, the symbol for °C is written so that the ° is underlined and separated from the letter C. Please correct this.
· All instruments used must be specified with the information about the model, the name of the manufacturer, the city and the state.
· Please write “Standard Method 4500-NH3 B” instead of “Standard Method 4500-NH3-B”.
· Please write T (sign for temperature) in italics.
· Page 3. “PO43–P (colorimetric method)” First, I assume that the minus after the number 3 is missing. Also, the authors stated that they used a colorimetric method, but no details of the method were given.
· Page 3. “The analysis of NO2-, NO3-, F-, SO42-, PO43-, Cl-, and Br- ions and Na+, B+, and Mg+ cations was conducted in accordance with the standard method using ionic chromatography…” First, the charge of the magnesium cation is 2+, and not +. Second, there is no information in the rest of the text about the analysis of B+ and Mg2+. Instead, the authors have provided information on the concentrations of K+ and Li+ ions (Table 1).
· In many places in the paper, the Latin letter “u” is used instead of the symbol “micro”. Please correct this.
· The name of package is “RStudio” and not “Rstudio”. Wouldn't it be easier to write once that RStudio version 1.2.1335 was used than to repeat it over and over again?
· Chapter Results, line 13 and Table 1 (twice). Correct PO42- to PO43-.
· Table 1. The range values and the average value should have the same number of digits.
· Table 1. Please replace “NO2–N” and “NO3–N” with “NO2––N” and “NO3––N”, respectively.
· Table 1. The average values are provided with standard deviations. However, what is the use of giving the standard deviation if some of these values are quite meaningless. For fluoride, for example, the standard deviation includes negative concentrations. What is the meaning of a negative concentration?
· Table 1. pH is not "potential of Hydrogen"!!! It is a measure of the concentration of hydrogen ions in the solution!
· Table 1. Is the detection limit for all measured quantities really 0.001? How did you determine this value? Furthermore, this value must not be given without the unit mg/L.
· Page 6. Figure 6b contains colors, but no legend explaining the meaning of those colors. I assume that the colors represent the values of the Pearson coefficient of determination. If so, what do the statements "is positively correlated (p > 0.7)" and "it is negatively correlated (p < 0.7)" mean?
· Text above Figure 3. “Finally, the average NH4+, Na+, NO2-, NO3-, Cl-, and SO42- removal efficiencies were 68.84%, 21.23%, -34.36%, -80.52%, 12.37%, and 33.36%, respectively.”
What do these minuses mean? Do they mean that the concentration of nitrates and nitrites has increased? If so, it should be said so, because the removal efficiency increases from 0 to 100%, and it is illogical to state this as a negative value.
· Figure 3. Numbers 2 and 3 (in NO2- and NO3-, respectively) are subscripted.
· Figure 3. Givet removal efficiencies as case b and not in the same diagram as the ion concentrations.
· Figure 4. Do not use the "equal to" sign. It sounds confusing when followed by < or >.
· Figure 5. It is not clear which bar refers to removal efficiency and which to concentration of coliforms, or percentage of resistance.
· What did you hope to get out of the PCA analysis? I do not see any important conclusion coming from the PCA analysis.
· “…which explains the variation in EC from 177.1 to 1704...” Shouldn’t these two values have the same number of decimal places?
· “There were also increases in bacteria resistant to the antibiotics CIP and CTX of -36.34% and -40.79%, respectively.” What is the meaning of these minuses?
· References. Check the references carefully and correct them according to the instructions for the authors. That is, do not write article number in parentheses after the volume number. Also, the volume number must be italicized. Use a semicolon instead of a comma to separate authors in the reference 13. Check the journal abbreviation in the reference 15. Period is missing in the journal abbreviations in references 44, 50 and 63. And so on.
The reviewer suggests major revision of the manuscript.
Author Response
Dear Reviewer 2,
The answers of your comments are in the attached file.
Thank you in advance.
Best regards,
Glaadys Vidal

Reviewer 3 Report
The amount of work presented in this manuscript is commendable. The authors investigated organic compounds and antibiotic-resistant bacterial behavior in CW-treated gray water, but there are still some shortcomings that must be resolved.
1. Why were cip, amx, and ctx chosen as ARB selection pressures?
2, Some of the results in Figure 3 are difficult to express intuitively.
3, The expression of rRNA 16s in 4.3 is incorrect.
4, 16S rRNA genes are DNA sequences corresponding to the coding rRNA of bacteria and are present in the genomes of all bacteria. Please explain why increasing coliform removals can increase ARB-removal efficiencies.
5, Please check all references one by one. Some references are mistaken in terms of the formatting of this journal.
Author Response
Dear Reviewer 3,
The answers of your comments are in the attached file.
Thank you in advance.
Best regards,
Glaadys Vidal

Round 2
Reviewer 2 Report
The reviewer suggests minor revision of the manuscript:
Table 1. The sulfate ion has the charge 2- and not 3-.
Author Response
Dear Reviewer 2,
Attached you can find a file with the response to your comments.
Thank in advance. With my best regards,
Gladys Vidal
